# Photodegradation of the H_1_ Antihistaminic Topical Drugs Emedastine, Epinastine, and Ketotifen and ROS Tests for Estimations of Their Potent Phototoxicity

**DOI:** 10.3390/pharmaceutics12060560

**Published:** 2020-06-17

**Authors:** Anna Gumieniczek, Anna Berecka-Rycerz, Urszula Hubicka, Paweł Żmudzki, Karolina Lejwoda, Paweł Kozyra

**Affiliations:** 1Department of Medicinal Chemistry, Medical University of Lublin, Jaczewskiego 4, 20-090 Lublin, Poland; anna.berecka@umlub.pl (A.B.-R.); k.lejwoda94@gmail.com (K.L.); pawekoz@interia.pl (P.K.); 2Department of Inorganic and Analytical Chemistry, Jagiellonian University, Collegium Medicum, Medyczna 9, 30-688 Cracow, Poland; urszula.hubicka@uj.edu.pl; 3Department of Medicinal Chemistry, Jagiellonian University, Collegium Medicum, Medyczna 9, 30-688 Cracow, Poland; zmudzki.p@gmail.com

**Keywords:** H_1_ antihistaminics, topical formulations, photodegradation kinetics, photodegradation pathways, phototoxicity, reactive oxygen species

## Abstract

In this study, important H_1_ antihistaminic drugs, i.e., emedastine (EME), epinastine (EPI), and ketotifen (KET), were irradiated with UV/Vis light (300–800 nm) in solutions of different pH values. Next, they were analyzed by new high performance liquid chromatography (HPLC) methods, in order to estimate the percentage of degradation and respective kinetics. Subsequently, ultra-performance liquid chromatography tandem-mass spectrometry (UPLC-MS/MS) was used to identify their photodegradation products and to propose degradation pathways. In addition, the peroxidation of linoleic acid and generation of singlet oxygen (SO) and superoxide anion (SA) were examined, together with the molar extinction coefficient (MEC) evaluation, to estimate their phototoxic risk. The photodegradation of all EME, EPI, and KET followed pseudo first-order kinetics. At pH values of 7.0 and 10.0, EPI was shown to be rather stable. However, its photostability was lower at pH 3.0. EME was shown to be photolabile in the whole range of pH values. In turn, KET was shown to be moderately labile at pH 3.0 and 7.0. However, it degraded completely in the buffer of pH 10.0. As a result, several photodegradation products were separated and identified using the UPLC-MS/MS method. Finally, our ROS assays showed a potent phototoxic risk in the following drug order: EPI < EME < KET. All of these results may be helpful for manufacturing, storing, and applying these substantial drugs, especially in their ocular formulations.

## 1. Introduction

The sensitivity of an active pharmaceutical substance (API) to light may vary with its chemical structure and reactivity, as well as the proposed dosage forms. For many APIs and formulations, photoreactivity leads to chemical degradation or to changes in their physical appearance [1,2]. Many classes of drugs also have the potential to provoke phototoxic, photoallergic, and photogenotoxic effects in tissues exposed to light [1,3].

Emedastine (EME), epinastine (EPI), and ketotifene (KET) belong to H_1_ histaminic receptor antagonists, exhibiting varied activity and several specific properties. EME is known as a competitive blocker of the H_1_ receptor, as well as an inhibitor of calcium ion influx across the basophil plasma membrane or intracellular calcium ion release [4]. EPI has properties which inhibit the release of mediators, such as histamine and leukotrienes. These include both antihistamine properties and chemical mediator stabilizer properties [4]. KET blocks the H_1_ receptor as a non-competitive blocker, stabilizes mast cells, inhibits the platelet activating factor, and acts as an eosinophil inhibitor [4,5]. The chemical structures of the drugs of interest are shown in Figure 1.

Blocking of the histamine H_1_ receptors serves as a primary therapeutic goal for topical anti-allergic medications. Therefore, EME, EPI, and KET, which are used in eye drops, external solutions, and ointments, can be mainly used to treat many allergic problems [5]. On the other hand, topically applied APIs may degrade upon extensive light exposure, lowering their therapeutic action or generating toxic products. In this context, the examination of their photodegradation constitutes a matter of great interest in the modern pharmacy field [6]. It is worth noting that photodegradation is not only related to changes in the structure of API, but also to the occurrence of free radical processes, energy transfer, or even luminescence, which may lead to unexpected results [7,8].

A review of older APIs in the area of photostability raises some concerns, as the data collected so far could be incomplete in the light of new guidelines. Moreover, the development of modern analytical methods has increased the possibility of the identification of new degradation products [7]. When it comes to studies on the stability of EME, EPI, and KET, the literary resources are rather limited. A literature survey reveals the presentation of a densitometry thin-layer chromatography (TLC) stability-indicating method in which the acidic and oxidative degradation of EME was described [9]. However, there is no data concerning the sensitivity to UV/Vis light exposure. In the literature concerning EPI, there are several HPLC methods for its determination in bulk drug substances, tablets, and ocular drops [10,11,12,13,14]. In these previous studies, EPI was degraded in acidic, alkaline, neutral, and oxidative conditions as a part of validation. In two studies [12,13], EPI was also exposed to UV light, but the breadth of the experiments was rather limited and no quantitative results were presented. A few HPLC stability-indicating methods have been presented for the determination of KET in different formulations [15,16,17,18]. However, degradation of the drug under UV/Vis irradiation has not been examined thus far and photodegradation kinetics has not been described.

Therefore, we carried out a detailed investigation on the photostability of EME, EPI, and KET under various pH conditions. Next, HPLC and UPLC-MS/MS experiments were performed for quantitative determinations and kinetic measurements, as well as for the separation and identification of respective photodegradation products. Since the literature does not contain any similar publications, the studies presented here are the first reports in this regard. In addition, the photochemical properties of EME, EPI, and KET were examined in different ways in terms of their phototoxicity. Firstly, we propose an approach involving UV/Vis measurements from 290 to 700 and determination of the molar extinction coefficient (MEC) values. Subsequently, the colorimetric monitoring of linoleic acid peroxidation, as well as singlet oxygen (SO) and superoxide anion (SA) generation in the presence of irradiated drugs, are reported.

## 2. Materials and Methods

### 2.1. Materials

The pharmaceutical grade standards of emedastine difumarate (EME), epinastine hydrochloride (EPI), ketotifen hydrogen fumarate (KET), chlorprotixene and todralazine hydrochlorides (internal standards for HPLC methods, i.s.), quinine hydrochloride and benzocaine (positive and negative controls for ROS determinations), imidazole, linoleic acid, p-nitrosodimethylaniline (RNO), nitroblue tetrazolium chloride (NBT), thiobarbituric acid (TBA), butylhydroxytoluene (BHT), 1,1′,3,3′-tetraethoxyoxypropane, 1-butanol, dimethyl sulphoxide (DMSO), and methanol from Merck (Darmstad, Germany); glacial acetic acid, sodium acetate, hydrochloric acid, sodium chloride, sodium tetraborate, phosphoric acid, sodium hydrogen phosphate, sodium dihydrogen phosphate, kalium dihydrogen phosphate, and kalium hydroxide from POCh (Gliwice, Poland); and formic acid, ammonium formate, acetonitrile, and water for LC/MS from J.T. Baker (Center Valley, PA, USA), were used.

The buffer solutions (acetate buffers of pH 3.0 and 4.8, phosphate buffers of pH 2.5 and 7.0, and borate buffer of pH 10.0) were prepared as described in European Pharmacopoeia [19]. The buffers of pH 3.0, 7.0, and 10.0 that were used as degradation media had the same ionic strength of 1 M, which was attained with 4 M sodium chloride. Sodium phosphate buffer of pH 7.4 (20 mM) that was used for the ROS assays was prepared according to respective literature [20].

### 2.2. Preliminary Spectrophotometric Measurements

EME, EPI, and KET were dissolved in methanol (1 mg/mL) and then diluted with 20 mM sodium phosphate buffer of pH 7.4 to a final concentration of 10 µg/mL. The absorption spectra in the range of 200–700 nm were recorded with a UV/Vis double beam spectrophotometer (CE 6600 from Cecil Instruments Ltd., Milton, UK) and a quartz cell with a 1 cm path length. The molar extinction coefficients (MEC) were determined from the absorbance values for the peaks at the maximum wavelengths.

### 2.3. Solar Simulator

The Suntest CPS Plus chamber from Atlas (Linsengericht, Germany) was used as a solar simulator irradiating UV and Vis light in the range of 300–800 nm, with the indicator setting value of 250 W/m^2^ for 1 h. The chamber was equipped with an appropriate temperature control unit and the temperature was adjusted at 25 °C during all experiments.

### 2.4. Photodegradation of EME, EPI, and KET in Solutions

Volumes of 1 mL from the stock solutions of EME, EPI, or KET (1 mg/mL) were dispensed to quartz glass-stoppered dishes and diluted with 1 mL of respective buffer solution (pH 3.0, 7.0, and 10.0). The samples were exposed to UV/Vis light with energies equal to 18,902, 37,804, 56,706, 75,608, and 94,510 kJ/m^2^. These doses of light were attained during 7, 14, 21, 28, and 35 h of irradiation, respectively. The starting energy of 18,902 kJ/m^2^ was equivalent to 1,200,000 lux·h and 200 W/m^2^, which is recommended by the ICH Q1B guidelines [21] as an initial dose of light that confirms drug stability, while the next doses were 2–5 times higher. After irradiation, the samples were diluted with methanol to cover the linearity range of our HPLC methods and analyzed quantitatively as described below. The assays were repeated three times for each sample (*n* = 3), and the concentrations of non-degraded drugs for each level of degradation (from 7 to 35 h of irradiation) were calculated from respective calibration equations and the starting concentration of EME, EPI, or KET.

#### Kinetics of Photodegradation

The concentrations of non-degraded drugs remaining after each time of irradiation or logarithms of these concentrations were plotted against the time of degradation to obtain the equation y = ax + b and the determination coefficient r, and in consequence, to determine the reaction order. Finally, further kinetic parameters, i.e., the degradation rate constant (k), degradation time of the 10% substance (t_0.1_), and degradation time of the 50% substance (t_0.5_), were calculated.

### 2.5. HPLC Method

#### 2.5.1. Chromatography

Analysis was performed with a model 515 pump, a Rheodyne 20 µL injector, and a model UV 2487 DAD, controlled by Empower 3 software, all from Waters UK Sales (Elstree, England). For EME and EPI, chromatography was carried out on a LiChrospher^®^100CN column (125 × 4.0 mm, 5 µm) from Merck. The mobile phase for EME was a mixture of methanol and phosphate buffer of pH 2.5 (40:60, *v*/*v*) with the flow rate of 2 mL/min. The UV detection was set at 280 nm. The mobile phase for EPI was the mixture of acetonitrile-methanol-acetate buffer of pH 4.8 (45:35:20, *v*/*v*/*v*) with the flow rate of 1.5 mL/min. The UV detection was set at 240 nm. For KET, chromatography was carried out on a LiChrospher^®^100RP-18 column (125 × 4.0 mm, 5 µm) from Merck. The mobile phase was a mixture of acetonitrile-methanol-acetate buffer of pH 4.8 (30:40:30, *v*/*v*/*v*) containing 0.25% formic acid with the flow rate of 2.0 mL/min, while the UV detection was set at 296 nm.

#### 2.5.2. Validation of the Methods

The selectivity of the methods was examined by determination of the stressed drugs in the presence of their degradation products. For calibration, working solutions of EME, EPI, or KET were prepared by dispensing 0.1–1.0 mL volumes from the stock solutions (1 mg/mL) to 10 mL volumetric flasks in order to reach the concentration range of 10–100 µg/mL. Internal standards (i.s.) were added as follows: 0.6 mL volumes of the stock solution of chlorprotixene (1 mg/mL) for EME, 0.2 mL volumes of the stock solution of todralazine (1 mg/mL) for EPI, and 0.5 mL volumes of chlorprotixene solution (10 mg/mL) for KET. After adjusting with methanol to the mark, solutions were injected into the column in five repetitions. The ratios of peak areas (the drug and respective i.s.) were plotted against the corresponding concentration of the drug to construct the calibration equations. The limit of detection (LOD) and the limit of quantification (LOQ) were determined from the SD of the intercept and the slope of the respective regression lines. In order to verify the accuracy and repeatability, replicate (*n* = 3) injections of working solutions of EME and EPI at low (15 µg/mL), medium (50 µg/mL), and high (90 µg/mL) concentrations were conducted. In the case of KET, the concentrations of 30, 50, and 70 µg/mL were injected into the column. The accuracy was calculated as the percentage of the analyte recovered by the respective assay, while repeatability was calculated as the relative standard deviation for the repeated between-day estimations.

### 2.6. UPLC-MS/MS Analysis

The UPLC-MS/MS system consisted of a Waters Acquity UPLC system coupled with a Waters TQD mass spectrometer (an electrospray ionization mode ESI-tandem quadrupole (Waters Corporation, Milford, MA, USA)). Chromatographic separations were carried out using the Acquity UPLC BEH C18 column (2.1 × 100 mm, 1.7 µm), equipped with the Acquity UPLC BEH C18 VanGuard pre-column (2.1 × 5 mm, 1.7 µm) from Waters. The column was maintained at 40 °C and eluted under gradient conditions using from 95% to 0% of eluent A over 10 min, at a flow rate of 0.3 mL/min. Eluent A was water/formic acid (0.1%, *v*/*v*) and eluent B was acetonitrile/formic acid (0.1%, *v*/*v*). Chromatograms were recorded using the Waters eλ PDA detector. Spectra were analyzed in the 200–700 nm range with a 1.2 nm resolution and sampling rate of 20 points/s. The MS detection settings of the Waters TQD mass spectrometer were as follows: source temperature, 150 °C; desolvation temperature, 350 °C; desolvation gas flow rate, 600 L/h; cone gas flow, 100 L/h; capillary potential, 3.00 kV; and cone potential, 30 V. Nitrogen was used for both nebulizing and drying gas. The data were obtained in a scan mode ranging from 50 to 1000 *m/z* at 0.5 s intervals; eight scans were summed up to get the final spectrum. Collision-activated dissociation (CAD) analyses were carried out with the energy of 50 eV. Consequently, the ion spectra were obtained by scanning from 50 to 500 *m/z*. The data acquisition software employed was MassLynx V 4.1 from Waters.

### 2.7. ROS Assays

For all ROS assays, EME, EPI, and KET were dissolved in a mixture of DMSO−20 mM sodium phosphate buffer of pH 7.4 (1:1, *v*/*v*) to obtain the 10 mM stock solutions. The indicator setting value of 250 W/m^2^ for 1 h was used for our CPS Plus chamber, achieving an intensity of UVA from 6.5 to 7.9 J/cm^2^. Our samples were irradiated for 15, 30, and 45 min and for 1, 3, and 7 h, which corresponded to energies equal to 675, 1350, 2025, 2700, 8100, and 18,902 kJ/m^2^.

Lipid peroxidation was measured using a TBA reactive substances (TBARs) test [22]. Briefly, linoleic acid was dissolved in 20 mM buffer of pH 7.4 (1 mM) and irradiated without or in the presence of EME, EPI, and KET (200 µM). Then, the irradiated samples (500 µL) were mixed with equal volumes of TBA (0.67% in 20 mM sodium phosphate buffer of pH 7.4) and 10 µL of BHT (1% in glacial acetic acid), and heated at 95 °C for 30 min. Finally, the samples were extracted with 1 mL of 1-butanol and centrifuged at 4000 rpm for 10 min, and the absorbance of the extracts was measured at 532 nm. The measurements were repeated three times (*n* = 3). A standard curve for TBARs was obtained with malondialdehyde (MDA) that had been generated by the hydrolysis of 1,1′,3,3′-tetraethoxypropane in acidic medium at a high temperature of 95 °C [22]. The calibration procedure with MDA as a standard was repeated five times (*n* = 5).

Singlet oxygen (SO) determination was based on bleaching p-nitrosodimethylaniline (RNO) using imidazole as a selective acceptor of SO [20,22,23]. The samples containing EME, EPI, and KET (200 µM) were mixed with RNO (50 µM) and imidazole (50 µM) in the quartz glass-stoppered dishes. The absorbance of each sample was measured at 440 nm. Then, the samples were exposed to UV/Vis light as described above, and the absorbance at 440 nm was measured again. In the superoxide anion test (SA), the reduction of nitroblue tetrazolium (NBT) was monitored as an increase in the absorbance at 560 nm [20,22,23]. The samples containing EME, EPI, and KET (200 µM) and NBT (50 µM) were mixed thoroughly, measured at 560 nm, exposed to UV/Vis light, and measured again.

All measurements were repeated three times for each sample (*n* = 3) and according to the results, the photoreactivity for EME, EPI, and KET was judged [20].

## 3. Results and Discussion

### 3.1. UV Measurements

As the primary event of a phototoxic reaction, the substance is excited by the absorption of photon energy, and the photoexcited chemical can then react with other substances through energy transfer and/or radical reactions, possibly leading to phototoxicity [3,22]. Therefore, a UV/Vis spectral analysis is recommended by the official guidelines as the first step of photosafety testing in the pharmaceutical industry [24]. A UV/Vis spectral analysis to clarify the potent photoreactivity of EME, EPI, and KET was carried out over the range of 200–700 nm. According to the spectral patterns of each drug, intense UV absorption was observed within the tested range (Figure 2). What is more, EME and KET showed at least one peak near or above 290 nm, partly overlapping with the sunlight spectrum, with MEC values greater than 10,000 L/(mol·cm). The UV absorption spectrum of EME was characterized by strong bands around 210, 255, and 290 nm. For the lowest energy band at 286 nm, the MEC value was calculated as 10,962 L/(mol·cm). EPI was shown to absorb appreciably over the range of 200–240 nm, while above 240 nm, the absorbance decreased gradually to 300 nm. The UV absorption spectrum of KET was characterized by two sharp peaks around 210 and 300 nm. For the lowest energy band at 296 nm, the MEC value was calculated as 15,617 L/(mol·cm). On the basis of these results, EME and KET were recognized as probably being excitative by sunlight and photoreactive, as well as potentially phototoxic [23].

### 3.2. Validation of the HPLC Method

Validation of our HPLC methods was performed with respect to the selectivity, linearity, LOD and LOQ, precision, and accuracy, according to ICH guidelines [25]. The respective validation parameters are shown in Table 1.

### 3.3. Percentage of Degradation of EME, EPI, and KET

The photodegradation of EME, EPI, and KET was estimated with our HPLC methods, after exposing the drugs to UV/Vis light with energies equal to 18,902, 37,804, 56,706, 75,608, and 94,510 kJ/m^2^ (exposure for 7, 14, 21, 28, and 35 h, respectively). As a result, EME was observed to be photolabile in a wide pH range, with a degradation value of 32.38–41.52% after the maximal irradiation. The chromatogram shows a decrease of the peak corresponding to the non-degraded EME (t_R_ 0.88 min). However, the appearance of any new peaks was not detected at 280 nm (Figure 3A). As far as EPI is concerned, the photodegradation was lower, achieving 8.10–19.10% with the pH decreasing from 10 to 3 after the highest irradiation (94,510 kJ/m^2^), without a corresponding rise in the peaks of degradation products in respective chromatograms (Figure 3B). When KET was irradiated with the maximal dose of light, it degraded to 14.64% at pH 3.0 and to 19.28% at pH 7.0. However, almost 100% degradation of KET was observed at pH 10.0 after a much lower dose of energy of 37,804 kJ/m^2^. The respective chromatogram shows the disappearance of the peak corresponding to unmodified KET (t_R_ 0.88 min), as well as the appearance of several new peaks of degradation products (Figure 3C).

It is well-known that the photodegradation of drugs is significantly affected by the pH of the environment because the changes in pH can promote or inhibit photolysis [26]. In addition, the intermediates and final products that are generated in the wake of these photochemical reactions may also change the pH of the environment. Moreover, the drug may be more stable in its ionized or non-ionized form and may undergo specific acid-base catalysis in aqueous solutions [26,27].

The drugs analyzed in this study are all weak bases with a pKa value of 8.68 for EME (the substituted diazepine ring), 8.77 for EPI (due to primary amine group substituted in the imidazo [1,5-a]azepine structure), and 8.43 for KET (the substituted piperidine ring) [28,29,30] (Figure 1). However, they differ as far as their percentage degradation in specific pH is concerned. Alkaline conditions showed a higher degradation efficiency and the maximum degradation for EME and KET was obtained at pH 10.0, where they were electrically neutral. On the other hand, a different trend in the case of EPI was observed and the maximum degradation occurred at pH 3.0, where its primary amine group was protonated (Table 2).

### 3.4. Kinetics of Degradation

Our study showed stronger correlations for the plots of logarithms of the concentration of non-degraded drugs than for the plots of the concentration of non-degraded drugs versus time of degradation, confirming pseudo first-order kinetics for their degradation, with all r values above 0.9. From these regression equations, further kinetic parameters were calculated, i.e., the degradation rate constant (k), degradation time of the 10% substance (t_0.1_), and degradation time of the 50% substance (t_0.5_) (Table 2).

As far as the photodegradation of EME is concerned, the calculated t_0.5_ values were 28.12 h at pH 3.0, 21.04 h at pH 7.0, and 19.92 h at pH 10.0, confirming its lowest stability in alkaline conditions. At the same time, EPI was shown to be less sensitive to UV/Vis light, with a percentage of degradation below 20%, even after the maximal dose of light. The calculated t_0.5_ values for EPI were 55.72 h at pH 3.0, 68.39 h at pH 7.0, and 107.5 h at pH 10.0, confirming its lowest stability in acidic conditions. As far as KET is concerned, the calculated t_0.5_ values dropped dramatically from 65.42 at pH 3.0 to 13.03 h at pH 7.0. What is more, the drug was completely degraded in pH 10.0, confirming its lowest stability in alkaline medium. The percentage degradation, as well as kinetic parameters for EME, EPI, and KET, are shown in Table 2. In turn, the pseudo first-order profiles of degradation of EME, EPI, and KET in solutions of different pH values are shown in Figure 4.

Because of the limited data from the literature concerning the photodegradation of EME, we could not conduct any detailed comparisons. As far as EPI is concerned, it was previously shown to be photostable in a solid state [13], whereas a 2% degradation value was observed for a methanolic solution [12]. However, no results on its degradation in solutions of different pH values and its degradation kinetics have been published so far. In terms of KET, only one similar study has been reported, in which KET was shown to be more stable than other benzocycloheptane antihistaminic agents [31]. In our study, KET was shown to be moderately sensitive to light at pH 3.0 and 7.0, and extremely sensitive at pH 10.0. Therefore, our results showed that EME and KET are sensitive to light in a wide pH range, which could be essential for manufacturing and protecting their formulations, especially in solutions. What is more, choosing the optimal excipients for their ocular drops becomes an important issue. As was described above, there are no articles in the literature concerning the kinetics of degradation with respect to EME, EPI, and KET. Therefore, the results presented here supplement the literary resources in this area.

### 3.5. LC/MS

The identification of photodegradation products (Ps) of EME, EPI, and KET was performed on the basis of our UPLC/MS analysis and supported with fragmentation patterns obtained from MS/MS experiments. A protonated molecule [M + H]^+^ of *m/z* 303.2 was observed in the full scan mass spectra of EME (Table 3). Its MS/MS showed the product ion at *m/z* 246.2 that followed a parallel fragmentation pathway to form ions at *m/z* 232.1 and 218.1. The subsequent step in the fragmentation pathway of EME was the loss of the water moiety from *m/z* 218.1, which resulted in the formation of a fragment with *m/z* 200.1. Following this, the last fragment formed an ion at *m/z* 174.1 that followed a parallel fragmentation to form two ions at *m/z* 146.1 and 134.1. All of these steps in the fragmentation pathway of EME are shown in Figure 5A. As was discussed above, EME was observed to be photolabile, with 32.38%, 38.34%, and 41.52% degradation at pH 3.0, 7.0, and 10.0, respectively. The degradation process at pH 3.0 was found to affect the ethoxy moiety, leading to the product EME-P1 with *m/z* 301.2 via hydroxylation and dehydration, while the fragment ions formed from EME P-1 were *m/z* 244.1, 218.1, and 146.1 (Figure 5B). In addition, the degradation at pH 3.0, 7.0, and 10.0 was shown to affect the 1,4-diazepine ring of EME, leading to its opening and to the product EME-P2 with *m/z* 277.2, with subsequent fragment ions at *m/z* 246.2, 218.1, 200.1, 174.1, 146.1, and 134.1 (Figure 5C). It is worth mentioning that the EME-P2 product was described in European Pharmacopoeia as Impurity F [19]. A respective UPLC chromatogram of EME and its degradation products is shown in Figure 6.

As far as EPI is concerned, a protonated molecule [M + H]^+^ of *m/z* 250.1 was observed in the full scan mass spectra (Table 4). Its MS/MS showed the product ion at *m/z* 208.1 that followed a parallel fragmentation to form ions at *m/z* 193.1 and 178.1. Next, the last fragment formed ions at *m/z* 130.1, 115.1, and 91.1. The subsequent steps in the fragmentation pathway of EPI, including the formation of other minor fragments, are depicted in Figure 7. The photodegradation of EPI achieved 8.10–19.10%, while the pH of the buffer decreased from 10 to 3. At pH 3.0, degradation was found to affect the imidazoline ring, leading to its oxidation, as was observed for the products EPI-P1, EPI-P2, and EPI-P3. The major ions formed during their MS/MS were *m/z* 248.1, 204.1, 178.1, and 127.1 (EPI-P1); 246.1, 204.1, and 178.1 (EPI-P2); and 204.1, 178.1, and 127.1 (EPI-P3). Further degradation of EPI led to imidazole ring opening, as was observed for the product EPI-P4 with the fragment ions at *m/z* 237.1, 210.1, and 192.1. It was interesting to observe that EPI-P3 and EPI-P4 were detected both at pH 3.0, where the highest degradation of EPI occurred (19.10%), as well as at pH 10.0, where degradation was relatively low (8.10%). At the same time, the EPI-P3 was described in European Pharmacopoeia as Impurity A [19]. The fragmentation patterns of EPI-Ps 1–4 are shown in Figure 8A–D, while their UPLC chromatogram is shown in Figure 9.

Lastly, KET showed a protonated molecule [M + H]^+^ at *m/z* 310.1 in the full scan mass spectra, while its MS/MS displayed product ions at *m/z* 249.0, 213.1, and 96.1 (Table 5). The subsequent steps in the fragmentation pathway of KET are shown in Figure 10. The drug was degraded in 14.64% at pH 3.0 and in 19.28% at pH 7.0. What is more, almost 100% degradation of KET was observed at pH 10.0. In all pH conditions degradation was found to affect the piperidine ring, leading to all of the identified products, i.e., KET-P1, KET-P2, and KET-P3, via its oxidation. Additionally, oxidation of the methyl substituent to a formyl moiety was observed at pH 3.0 (KET-P2) and demethylation of the nitrogen atom of the piperidine moiety was observed at pH 7.0 (KET-P3). The major ions formed during their MS/MS were *m/z* 324.1, 306.1, 278.1, and 221.0 (KET-P1); 292.1, 263.2, 275.1, and 221.0 (KET-P2); and 324.1, 306.1 278.1, 250.1, and 221.0 (KET-P3). The respective fragmentation patterns and corresponding UPLC chromatogram are shown in Figure 11, Figure 12, Figure 13 and Figure 14.

### 3.6. ROS Assays

In the present experiments, the samples of EME, EPI, and KET were irradiated for 15, 30, and 45 min, and for 1, 3, and 7 h, which corresponded to energies equal to 675, 1350, 2025, 2700, 8100, and 18,902 kJ/m^2^. Bearing in mind that the indicator setting value of 250 W/m^2^ for 1 h achieved an intensity of UVA from 6.5 to 7.9 J/cm^2^ for our CPS Plus chamber, the selected light energies comprised the range of 5–20 J/cm^2^, which is recommended in all phototoxicity assays [20,22,23]. To better verify our results, quinine and benzocaine were included in our ROS experiments as a positive and negative control, respectively.

As a starting point to evaluate the phototoxic potential of EME, EPI, and KET, the photosensitized lipid peroxidation was monitored using linoleic acid as a substrate. It is known from the literature that phototoxic drugs accelerate the peroxidation of lipids upon exposure to UV/Vis radiation [22]. In addition, lipid peroxidation in the cellular membrane is considered to be one of the major mechanisms in drug-induced photoirritation [22].

In the present study, lipid peroxidation was monitored as TBARs determination with MDA as a standard. The linearity between the level of MDA and the absorbance at 532 nm due to the product of MDA with TBA was investigated. The regression line equation calculated by the least squares method was y = 1.1287 ± 0.0056x + 0.0261 ± 0.0011, with a coefficient of correlation of r = 0.9997 ± 0.0005 (mean ± SD, *n* = 5). On the basis of the above calibration equation, the number of TBARs generated under the irradiation of linoleic acid in the absence or presence of EME, EPI, and KET was determined. It was clearly shown that the peroxidation of linoleic acid was at the level of the control group when the drugs were absent. However, irradiation of the substrate in the presence of EME and KET produced a much larger number of TBARs. What is more, the results obtained for KET were similar to those of quinine, which is known to be a highly photoreactive substance. In contrast, the addition of EPI to linoleic acid increased peroxidation to a lesser extent, although the content of TBARs was higher than in the presence of benzocaine (a negative control) (Table 6). At the same time, the obtained results were correlated with high values of MEC for EME and KET at wavelengths near 290 nm. Therefore, EME and especially KET can be considered to have a potent photoirritancy in vitro, on the basis of their ability to enhance the peroxidation of linoleic acid [22].

According to the official guidelines [20,22], the phototoxic potential of drugs can be elucidated by monitoring ROS generation under exposure to UV/Vis radiation, as SO (type 2 photochemical reaction) and SA (type 1 photochemical reaction) generation. In our experiments, each drug was judged to be photoreactive when (a) SO values of 25 or more and SA values of 70 or more were measured; (b) SO values of less than 25, but SA values of 70 or more, were measured; and (c) SO values of 25 or more and SA values of less than 70 were measured. When SO values were lower than 25 and SA values were between 20 and 70, the drug was judged as weakly photoreactive [22]. According to these criteria and the standard intensity of UVA from 6.5 to 7.9 J/cm^2^, corresponding to 2700 kJ/m^2^_,_ EME and KET were identified as phototoxic drugs by mechanisms including ROS generation (SO > 25 and SA > 70). In contrast, EPI was judged as weakly photoreactive (SO < 25 and 20 > SA < 70) (Table 6). Therefore, EME and KET would have substantial photoreactivity and could potentially cause phototoxic reactions in vivo, upon exposure to sunlight. The obtained results clearly show the necessity for further experiments concerning the phototoxicity of EME and KET, especially considering that they are more vulnerable to degradation when applied topically [8].

Almost all antihistamines have been reported in the literature as causing hypersensitivity reactions, including their topical formulations. Shakouri and Bahna [32] published a comprehensive review on antihistaminic allergies reported between 1949 and 2013. More than one hundred cases were identified, which included urticaria, angioedema, anaphylaxis, fixed drug eruptions, allergic contact, and photosensitive dermatitis, as well as generalized nonspecific rashes [32,33]. Therefore, our observations in vitro could be in agreement with the phototoxicity data from these clinical studies [33].

Among different types of products, the stability of ophthalmic pharmaceuticals is the most important, since eyes are very sensitive. They are easily irritated if the composition of the ophthalmic pharmaceutical is not adequate [34]. Therefore, ophthalmic pharmaceuticals must be suitably compounded and packaged for their application to the eye [5]. The pH value is often the stability-controlling factor for many products, but it must be within a certain range because of the bioavailability of the drug [34]. Our results clearly show the need for optimal pH selection for EME and KET, bearing in mind both of the questions presented above.

## 4. Conclusions

The present study was designed and performed considering the importance of EME, EPI, and KET in therapy and their topical use as ocular drops, as well as the lack of reports concerning their degradation, isolation, and characterization of their degradation products. As a result, we presented complete photostability data for EME, EPI, and KET at different pH values, their kinetics of degradation, and characterization of their main photodegradation products. Several degradation products of EME, EPI, and KET were detected and identified by our UPLC-MS method that have not previously been described in the literature, i.e., EME-P1; EPI-Ps 1, 2, and 4; and KET-Ps 1–3.

In addition, EME and KET were deduced to have a phototoxic risk in UV/Vis absorption measurements and their MEC values. Because of several false negative predictions based on the MEC values which were reported in the literature, a combined use of other photosafety testing is recommended to ensure the photosafety of drugs [20,21]. As a consequence, the MEC evaluation, ROS assays, and photosensitized peroxidation of linoleic acid were reported. Therefore, the results developed here can be utilized to improve the quality of topical formulations of EME, EPI, and KET. They can also be used to minimize the risk associated with their photodegradation, as well as the phototoxic risk of EME and KET in terms of patients’ health.

## Figures and Tables

**Figure 1 pharmaceutics-12-00560-f001:**
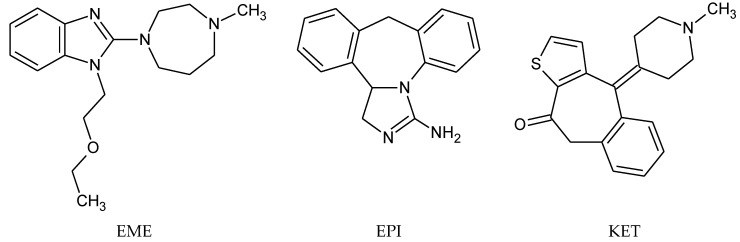
Chemical structures of emedastine (EME), epinastine (EPI), and ketotifen (KET).

**Figure 2 pharmaceutics-12-00560-f002:**
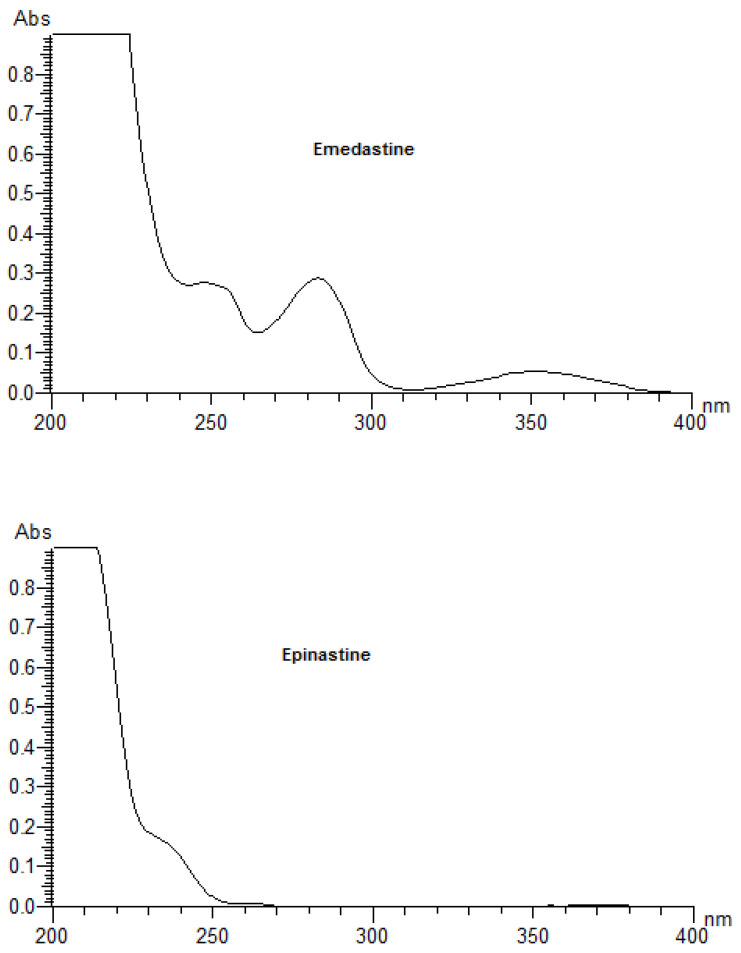
UV spectra of the examined drugs (concentration of 0.01 mg/mL in methanol).

**Figure 3 pharmaceutics-12-00560-f003:**
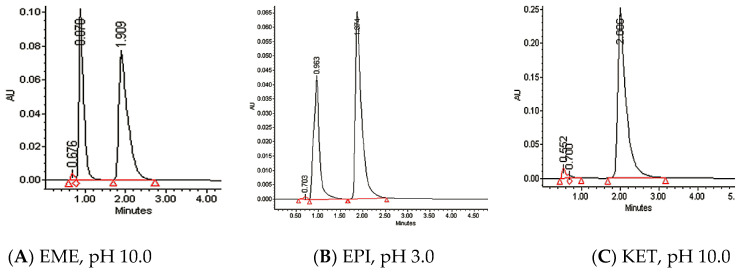
Chromatograms of (**A**) emedastine (EME), (**B**) epinastine (EPI), and (**C**) ketotifen (KET) after UV/Vis irradiation in buffer solutions.

**Figure 4 pharmaceutics-12-00560-f004:**
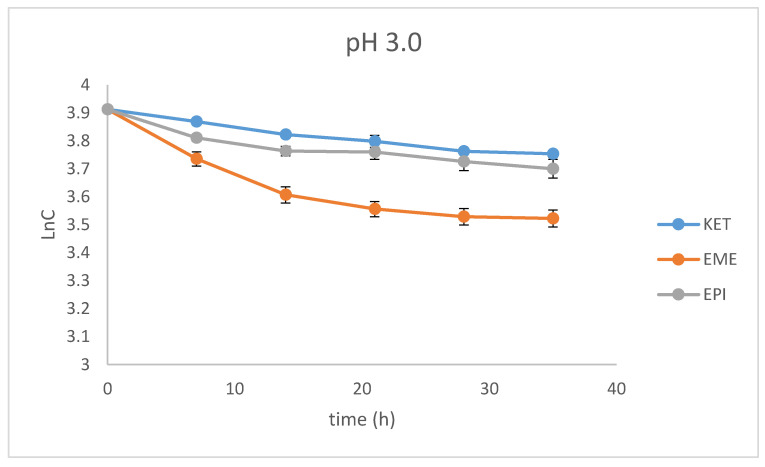
First-order plots of the degradation of emedastine (EME), epinastine (EPI), and ketotifen (KET) in buffer solutions (*n* = 3, mean ± SD).

**Figure 5 pharmaceutics-12-00560-f005:**
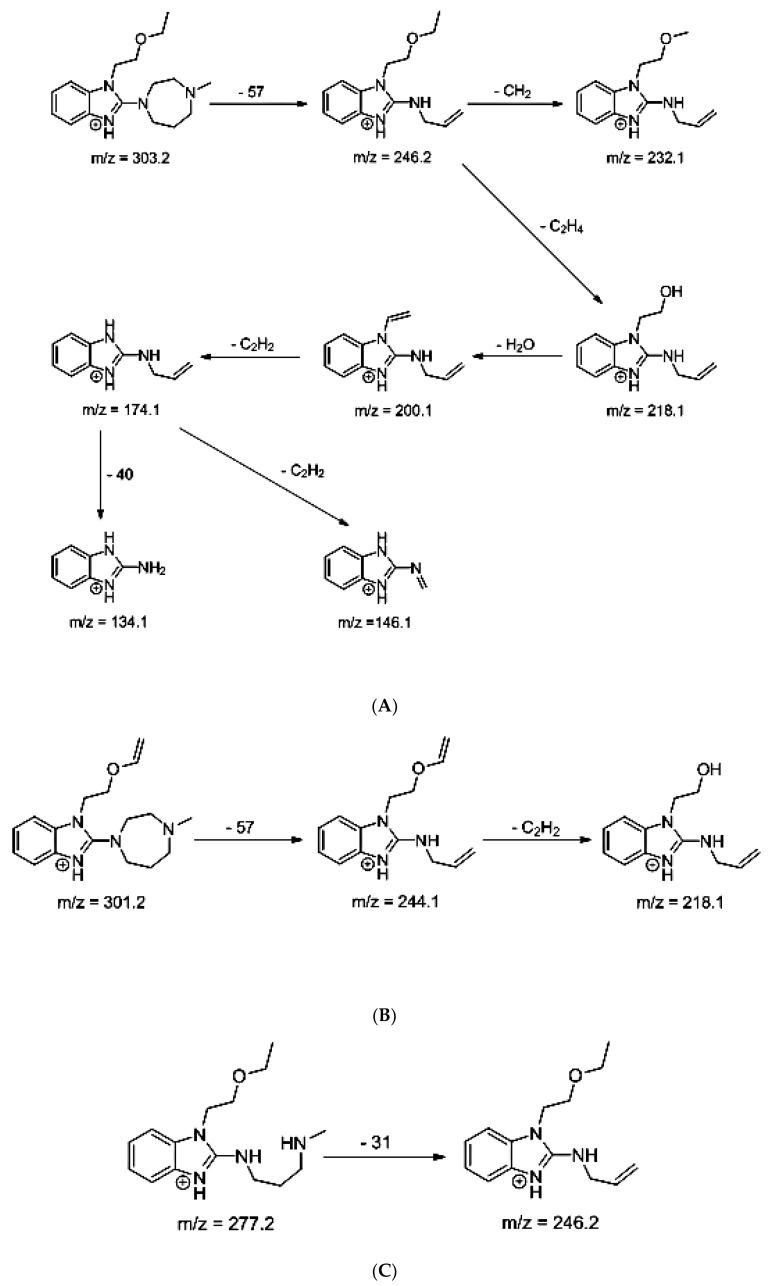
Proposed fragmentation patterns of emedastine (EME) (**A**), EME-P1 (**B**), and EME-P2 (**C**).

**Figure 6 pharmaceutics-12-00560-f006:**
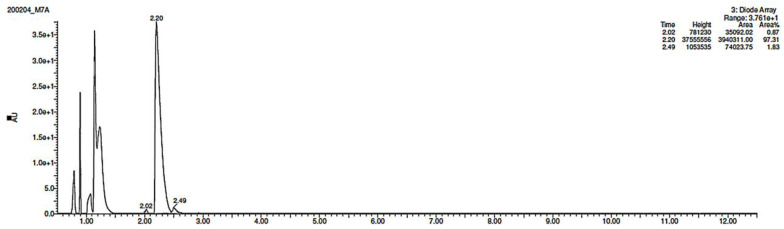
UPLC chromatogram of emedastine (EME) and its degradation products in a solution of pH 3.0.

**Figure 7 pharmaceutics-12-00560-f007:**
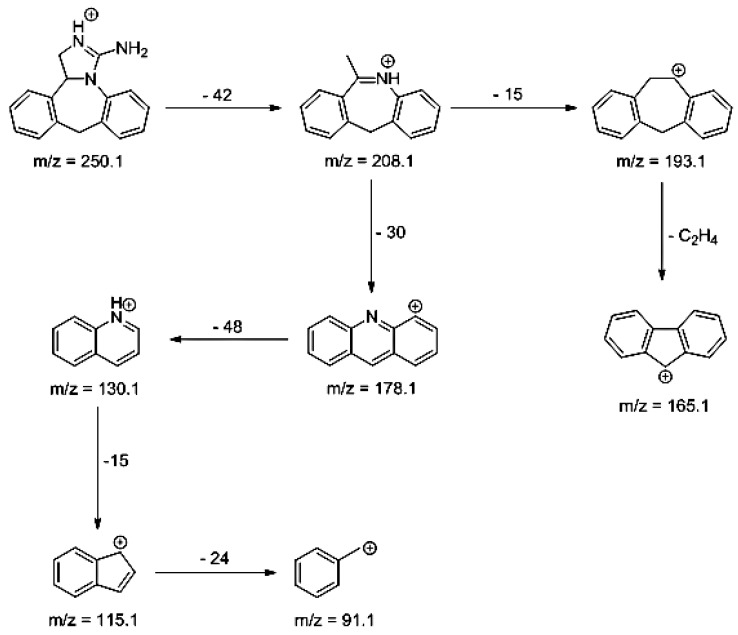
Proposed fragmentation pattern of epinastine (EPI).

**Figure 8 pharmaceutics-12-00560-f008:**
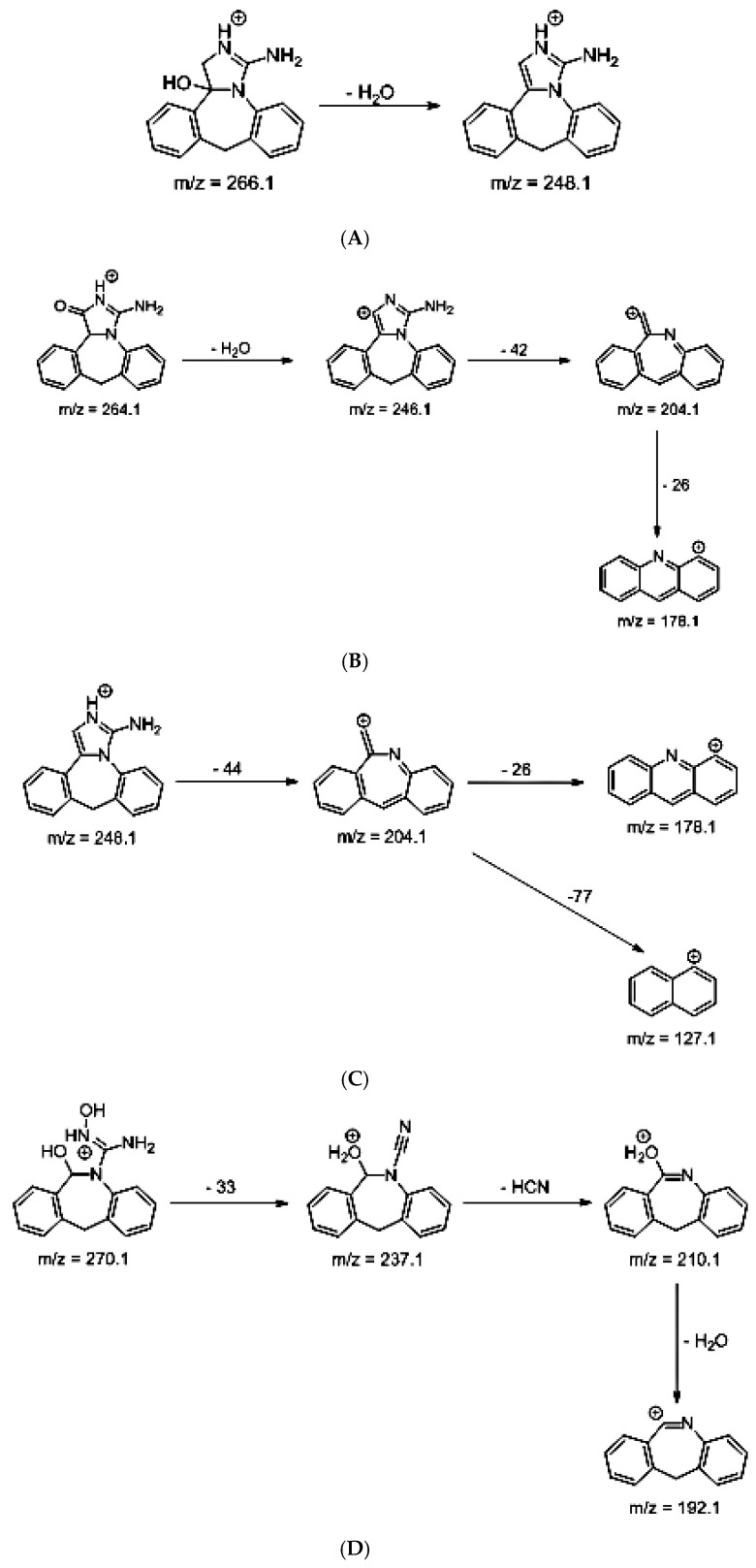
Proposed fragmentation patterns of EPI-P1 (**A**), EPI-P2 (**B**), EPI-P3 (**C**), and EPI-P4 (**D**).

**Figure 9 pharmaceutics-12-00560-f009:**
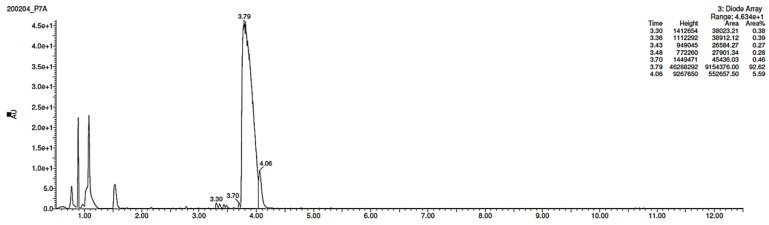
UPLC chromatogram of epinastine (EPI) and its degradation products in a solution of pH 3.0.

**Figure 10 pharmaceutics-12-00560-f010:**
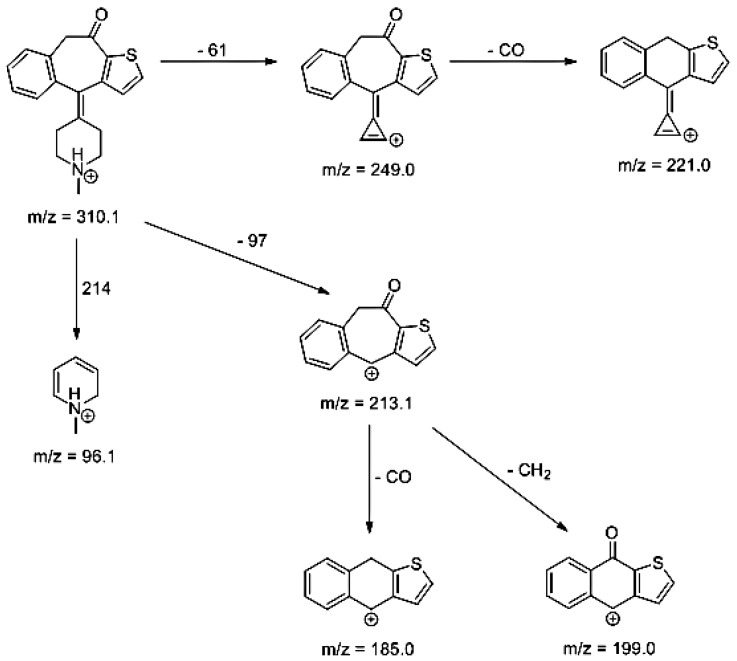
Proposed fragmentation pattern of ketotifen (KET).

**Figure 11 pharmaceutics-12-00560-f011:**
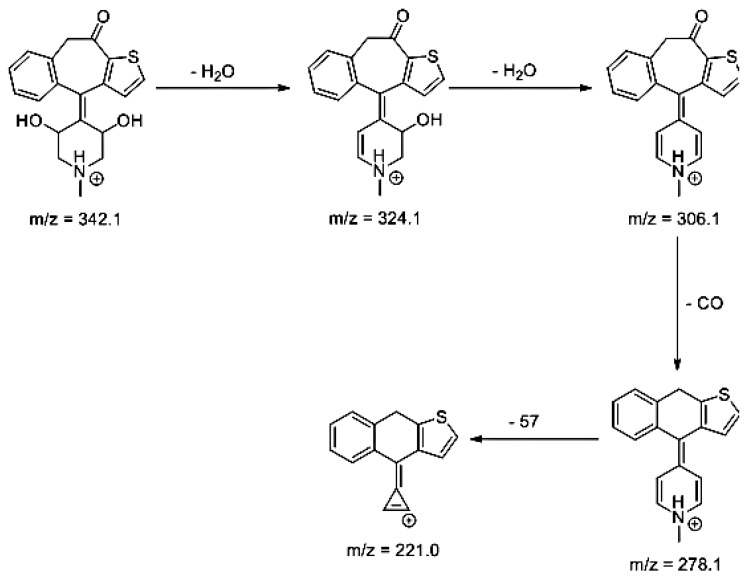
Proposed fragmentation pattern of KET-P1.

**Figure 12 pharmaceutics-12-00560-f012:**
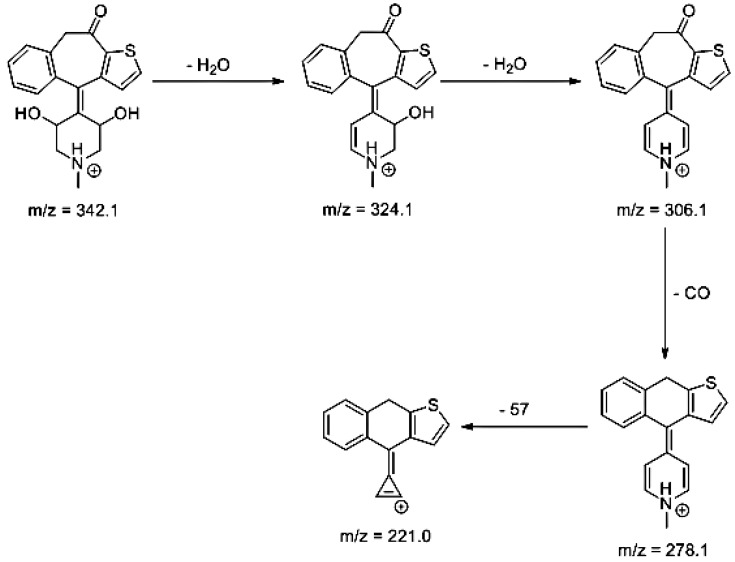
Proposed fragmentation pattern of KET-P2.

**Figure 13 pharmaceutics-12-00560-f013:**
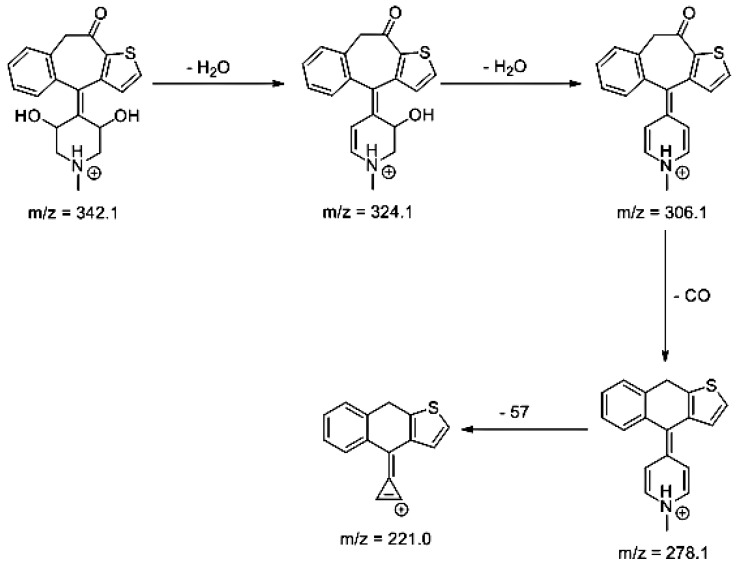
Proposed fragmentation pattern of KET-P3.

**Figure 14 pharmaceutics-12-00560-f014:**
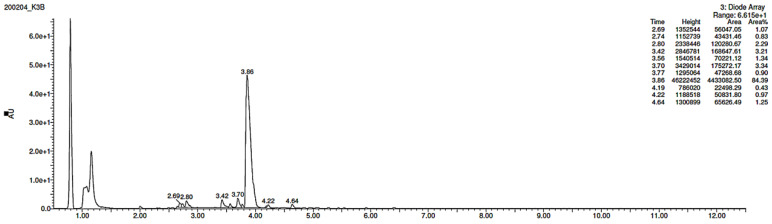
UPLC chromatogram of ketotifen (KET) and its degradation products in a solution of pH 7.0.

**Table 1 pharmaceutics-12-00560-t001:** Validation of HPLC methods for the determination of emedastine (EME), epinastine (EPI), and ketotifen (KET) (*n* = 5).

Parameter	Values		
	EME	EPI	KET
Linearity range (µg/mL)	10–100	10–100	10–100
Slope	0.01942	0.03020	0.0061
SD of the slope	0.00129	0.00028	0.00002
Intercept	0.06199	0.04752	0.0018
SD of the intercept	0.00913	0.00475	0.00012
r^2^	0.9991	0.9962	0.9997
SD of the r^2^	0.00011	0.00042	0.00018
LOD (µg/mL)	1.41	0.47	0.06
LOQ (µg/mL)	4.70	1.57	0.20
Accuracy (% Recovery)	97.95	104.65	102.69
SD of the recovery	1.73	1.14	2.10
Inter day precision (% RSD)	2.06	1.03	0.21

**Table 2 pharmaceutics-12-00560-t002:** Kinetics of the photodegradation of emedastine (EME), epinastine (EPI), and ketotifen (KET) in buffer solutions (*n* = 3).

Conditions	Degradation (%)	y = ax + b	r	K (min^−1^)	t_0.1_ (h)	t_0.5_ (h)
**EME**						
Buffer pH 3.0	32.38	y = −0.0107x + 3.8306	0.9136	4.11 × 10^−4^	4.26	28.12
Buffer pH 7.0	38.34	y = −0.0143x + 3.8604	0.9390	5.48 × 10^−4^	3.19	21.04
Buffer pH 10.0	41.52	y = −0.0151x + 3.8371	0.9543	5.79 × 10^−4^	3.02	19.92
**EPI**						
Buffer pH 3.0	19.10	y = −0.0054x + 3.8728	0.9351	2.07 × 10^−4^	8.44	55.72
Buffer pH 7.0	12.16	y = −0.0044x + 3.9112	0.9860	1.69 × 10^−4^	10.4	68.39
Buffer pH 10.0	8.10	y = −0.0028x + 3.9214	0.9604	1.07 × 10^−4^	16.3	107.5
**KET**						
Buffer pH 3.0	14.64	y = −0.0046x + 3.9004	0.9823	1.77 × 10^−4^	9.91	65.42
Buffer pH 7.0	19.28	y = −0.0049x + 3.8804	0.9303	8.87 × 10^−4^	1.97	13.03
Buffer pH 10.0	100	-	-	-	-	-

**Table 3 pharmaceutics-12-00560-t003:** Proposed structures of the photodegradation products of emedastine (EME).

Compound	t_R_ (min)	[M + H] ^+^	Fragmentation Ions	Structure
EME	2.20	303.2	134.1, 146.1, 174.1, 200.1, 218.1, 232.1, 246.2	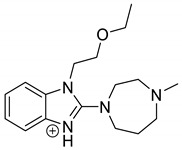
EME-P1 pH 3.0	2.02	301.2	146.1, 218.1, 244.1	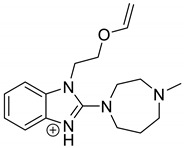
EME-P2 = Impurity F [19] pH 3.0 pH 7.0 pH 10.0	2.49	277.2	134.1, 146.1, 174.1, 200.1, 218.1, 246.2	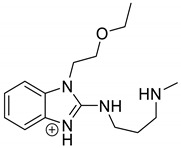

**Table 4 pharmaceutics-12-00560-t004:** Proposed structures of the photodegradation products of epinastine (EPI).

Compound	t_R_ (min)	[M + H] ^+^	Fragmentation Ions	Structure
EPI	3.79	250.1	91.1, 115.1, 130.1, 165.1, 178.1, 193.1, 208.1	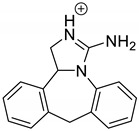
EPI-P1 pH 3.0	3.30	266.1	127.1, 178.1, 204.1, 248.1	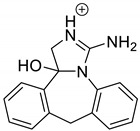
EPI-P2 pH 3.0	3.43	264.1	178.1, 204.1, 246.1	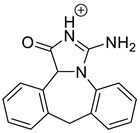
EPI-P3 = Impurity A [19] pH 3.0 pH 10.0	4.06	248.1	127.1, 178.1, 204.1	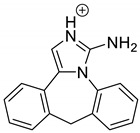
EPI-P4 pH 3.0 pH 10.0	3.35	270.1	192.1, 210.1, 237.1	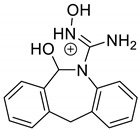

**Table 5 pharmaceutics-12-00560-t005:** Proposed structures of the photodegradation products of ketotifen (KET).

Compound	t_R_ (min)	[M + H] ^+^	Fragmentation Ions	Structure
KET	3.79	310.1	96.1, 185.0, 199.0, 213.1, 221.0, 249.0	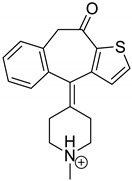
KET-P1 pH 7.0	3.43	342.1	324.1, 306.1, 278.1, 221.0	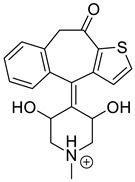
KET-P2 pH 7.0	3.56	320.1	247.1, 275.1, 263.1, 292.1	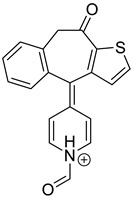
KET-P3 pH 7.0	3.69	342.1	221.0, 250.1, 278.1, 306.1, 324.1	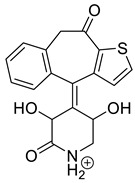

**Table 6 pharmaceutics-12-00560-t006:** Results of the ROS tests of emedastine (EME), epinastine (EPI), and ketotifen (KET) versus quinine as a positive control and benzocaine as a negative control (*n* = 3).

Energy (kJ/m^2^)	EME	EPI	KET	Quinine	Benzocaine
**TBARs**					
675	0.107	0.087	0.120	0.162	0.055
1350	0.118	0.089	0.121	0.170	0.057
2025	0.122	0.091	0.136	0.175	0.062
2700	0.131	0.093	0.141	0.189	0.063
8100	0.133	0.101	0.207	0.244	0.078
18,902	0.136	0.109	0.274	0.285	0.081
**SO**					
675	8	6	9	282	−9
1350	10	8	11	319	−8
2025	21	8	27	401	−3
2700	37	23	55	493	10
8100	72	27	173	543	13
18,902	101	35	452	586	21
**SA**					
675	29	6	60	133	−11
1350	34	21	86	148	−8
2025	49	29	110	179	−6
2700	110	34	151	221	2
8100	216	44	358	422	8
18,902	226	62	424	540	9

TBARs, thiobarbituric acid reactive substances test; SO, singlet oxygen test; SA, superoxide anion test.

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
