# Peer review of "Photodegradation of the H1 Antihistaminic Topical Drugs Emedastine, Epinastine, and Ketotifen and ROS Tests for Estimations of Their Potent Phototoxicity"

_pharmaceutics, 2020, doi:10.3390/pharmaceutics12060560_

Round 1

Reviewer 1 Report

In this study, Gumieniczek et al., have shown in vitro the photostability of emedastine (EME), epinastine (EPI) and ketotifen (KET) under different pH conditions identifying several new photodegradation products by different biochemical approaches such as HPLC, UPLC-MS/MS and ROS assay.

Overall, the paper is very interesting to minimize the risk associated to phototoxic effects of drugs as in the case of EME, EPI and KET. The experimental design appears simple but very appropriate and the results are in accordance with the conclusions.

However, there are major and minor points (detailed below) to be resolved before considering it suitable for publication:

Major Points:

  • The most important limitation of this study is the analysis reported only in vitro experiments. A large number of considerations that the authors discuss must be confirmed at least by cellular experiments:
    1. These molecules could be considered as having a potent photoirritancy in vitro based of their ability for enhancing peroxidation of linoleic acid by ROS production. The authors must confirm these effects on a cellular model.
    2. Different degradation products of EME, EPI and KET not described in the literature were identified in this study. What are the toxic effects of these molecules?
    3. For these new degradation products the authors must show the toxic effects in cellular experiments.

Minor Points:

  • Many affirmations and literature data are lacking of specific references. The authors must insert them.
  • In materials and methods for each tests used the number of the experiment must be described.
  • In the Figure 2, for each points no error bars are reported.

Author Response

The most important limitation of this study is the analysis reported only in vitro experiments. A large number of considerations that the authors discuss must be confirmed at least by cellular experiments:

    1. These molecules could be considered as having a potent photoirritancy in vitro based of their ability for enhancing peroxidation of linoleic acid by ROS production. The authors must confirm these effects on a cellular model.
    2. Different degradation products of EME, EPI and KET not described in the literature were identified in this study. What are the toxic effects of these molecules?
    3. For these new degradation products the authors must show the toxic effects in cellular experiments.

Thank you for all these comments. The three points above emphasize the need to confirm the phototoxic potential of emedastine and ketotifen in cellular experiments. We agree that all phototoxic effects should be confirmed in experiments on appropriate cell lines. The goal of the presented study, however, was to conduct in vitro tests, mainly to compare photosensitivity of three examined drugs, i.e. emedastine, epinastine and ketotifen, to examine their photodegradation under different conditions and to detect any “chemical” reasons for further studies on their potential phototoxicity. Presently, cellular experiments are being planned, e.g. 3T3 NRU-PT test, MTT test and comet assay to estimate phototoxicity and potential photogenotoxicity. In the course of carrying out these planned tests, the observations described in this work will be used, also in terms of optimal conditions for further experiments. We are currently preparing such experiments, but they require an extension our team to include a specialist in cellular research. It is not possible for us to perform such tests in the near future.

Minor Points:

  • Many affirmations and literature data are lacking of specific references. The authors must insert them.

Respective references were inserted in our corrected text (lines: 35, 41, 53, 58, 205, 218, 245, 383, 399, 405, 418, 424, 428, 429).

Additionally, 2 new references were added (lines: 462, 469).

  • In materials and methods for each tests used the number of the experiment must be described.

The number of the experiments were added everywhere that is concerned, i.e. in paragraphs 2.4, 2.6 and 2.7 (lines: 119, 151, 152, 186, 189, 199, 259, 402).

  • In the Figure 2, for each points no error bars are reported.

Error bars (mean±SD) were added for each point of time (line 276).

Reviewer 2 Report

The manuscript “Photodegradation of H1 antihistaminic topical drugs, emedastine, epinastine and ketotifen, and ROS tests for estimation of their potent phototoxicity” is a comprehensive study of the photodegradation of three commercial anti-histaminic topical drugs. Gumieniczek and co-authors unveiled new insights on the photodegradation products, degradation pathways, kinetics degradation, and potential phototoxicity risk of emedastine, epinastine, and ketotifen active pharmaceutical substances.

The manuscript is written clearly and logically and the study was well conducted. The photodegradation protocol followed the recommendations of ICH Q1D guidelines (Guideline for Stability Testing of New Active Substances and Medicinal Products) where the experiments were taken in buffer solutions with 3 different pH ranges (3, 7 and 10) and under different light doses. The phototoxic risk, for each drug, has been accessed by using the peroxidation of linoleic acid as a probe. The photodegradation yields and kinetics studies have been achieved by HPLC methods, whilst the photodegradation pathways and photodegradation products have been proposed by means of UPLC-MS/MS. The results of this study undoubtedly are interesting for researchers working in various areas of drug development. The outcome of the present study might be helpful to better understand and mitigate the risks associated with the pharmaceutical development and storage of topical formulations of EME, EPI, and KET drug products. The conclusions are in accordance with the results presented. I believe the manuscript is a valuable scientific contribution on the field, and adequate for the area of interests of the journal Pharmaceutics.

Minor concerns:

-The graphical quality of chromatograms needs improvement.

-In order to support the discussion of section 3.1 UV measurements, it would be interesting to insert the Uv-Vis absorption spectra of the 3 molecules under study. It would be also useful and informative for readers to insert the EME, EPI and KET chemical structures. Please provide and include the aforementioned information. 

-page 6, line 248: The term “protonized” sounds weird.

Author Response

  • -The graphical quality of chromatograms needs improvement.

The quality of chromatograms was improved (lines: 316, 342, 368).

  • -In order to support the discussion of section 1 UV measurements, it would be interesting to insert the UV-Vis absorption spectra of the 3 molecules under study. It would be also useful and informative for readers to insert the EME, EPI and KET chemical structures. Please provide and include the aforementioned information. 

The UV/Vis spectra of emedastine, epinastine and ketotifen was added in a new Figure 2 (line 45).

The chemical structures of emedastine, epinastine and ketotifen were added in a new Figure 1 (line 219).

  • -page 6, line 248: The term “protonized” sounds weird.

Thank you for this suggestion. It was corrected (line 256).

Thank you very much for all suggestions.